

# Effects of bilateral low-load resistance training with blood-flow restriction on post-activation performance enhancement in male collegiate athletes

Zhanfei Zheng[1,2], Zhipeng Wang[1], Changyuan Duan[1], Yu Zhang[1], Xiaowei Wang[1], Xixi Miao[3] and Haiping Liu[1]

[1] Department of Sports Science, Wenzhou Medical University, Wenzhou, Zhejiang, China
[2] School of Strength and Conditioning Training, Beijing Sport University, Beijing, China
[3] Wenzhou Nanpu Experimental Middle School, Wenzhou, Zhejiang, China

Corresponding author
Haiping Liu, lhp@wmu.edu.cn

## ABSTRACT

**Background**. While the effects of blood-flow restriction (BFR) training on various performance outcomes have been widely studied, the combination of BFR with low-load (LL) resistance training for post-activation performance enhancement (PAPE), particularly in vertical jump performance, has not been fully explored. This study aimed to investigate whether combining BFR with LL resistance training can enhance vertical jump performance in male collegiate athletes.

**Methods**. Fifteen male strength trainers (mean $\pm$ standard deviation (SD): age 21.7 $\pm$ 1.4 years, body mass 77.2 $\pm$ 6.3 kg, and height 179.1 $\pm$ 5.7) with at least two years of resistance training experience participated in three experimental trials using a randomized crossover design with 72-hour intervals: (a) low-load resistance exercise at 30% one repetition maximum (1RM) back squat combined with BFR for four sets of 15 repetitions (BFR+LL); (b) low-load resistance exercise without BFR for 4 sets of 15 repetitions (LL); and (c) a control condition involving passive rest (CON). Countermovement jump (CMJ) performance , including vertical jump height (VJH), relative peak power output (RPP), force impulse (FI), and rate of force development (RFD), was assessed at baseline, immediately after, and at 3-, 6-, 9-, and 12-minutes post-protocol. Both peak and mean CMJ were measured to evaluate performance changes.

**Results**. The repeated measures analysis of variance (ANOVA) revealed significant condition $\times$ time interactions ($p < 0.001$) for VJH, RPP, and FI. Post-hoc comparisons demonstrated that BFR+LL resulted in significant improvements in VJH, RPP, and FI at 12 minutes post-protocol relative to both the CON and LL conditions ($p < 0.05$). Specifically, VJH increased by 7.17% (effect size (ES) = 0.79), RPP by 2.26% (ES = 0.31), and FI by 3.21% (ES = 0.29) compared to CON at 12 minutes following the BFR+LL protocol. In contrast, a significant decline in performance ($p < 0.05$) was observed immediately after BFR+LL, with VJH decreasing by $-9.1 \pm 5.1\%$ (ES = $-1.12$), RPP by $-8.3 \pm 4.3\%$ (ES = $-1.16$), and FI by $-5.0 \pm 2.2\%$ (ES = $-0.44$) compared to baseline. No significant changes in RFD or peak CMJ performance were observed across the three conditions ($p > 0.05$).

**Conclusion**. The study suggests that BFR+LL resistance training may enhance acute vertical jump performance 12 minutes post-exercise, despite an initial decline in performance immediately following the protocol.

## INTRODUCTION

Post-activation performance enhancement (PAPE) is a practical strategy used to enhance muscular strength or power performance following the completion of conditioning activity (CA) in various sports, including athletic competitions and ball games (*Dello Iacono, Martone & Padulo, 2016*; *Till & Cooke, 2009*; *Tsoukos et al., 2016*). Previous research suggests that the potential mechanisms of PAPE may involve phosphorylation of myosin light chains (*Hodgson, Docherty & Robbins, 2005*; *Smith & Fry, 2007*), increased motor unit recruitment (*Sale, 2002*), and alterations in the angle of muscle fiber pennation (*Blazevich & Babault, 2019*). These physiological changes enhance force generation and power output. Although the exact biological processes remain incompletely understood, achieving an optimal PAPE effect requires a balance between fatigue and potentiation during conditioning activity (*Jami et al., 1983*; *Rankin et al., 1988*; *Rassier & Macintosh, 2000*; *Vergara, Rapoprot & Nassar-Gentina, 1977*). This balance is influenced by factors such as the individual's physiological characteristics (*e.g.*, muscle fiber composition, neuromuscular adaptations, and recovery capacity) (*Chiu et al., 2003*), the intensity and volume of conditioning activity, rest intervals (*Wilson et al., 2013*), and the athlete's training status (*Suchomel et al., 2016*).

Resistance training (RT) protocols involving 1–5 repetitions at 80–90% of one repetition maximum (1RM) are commonly used to trigger PAPE (*Seitz & Haff, 2016*; *Wilson et al., 2013*). However, high-load (HL) exercises can lead to significant mechanical strain on the lower extremities making it unsuitable for athletes undergoing multiple training sessions per day or individuals undergoing rehabilitation (*Haff & Triplett, 2016*). These individuals typically exhibit lower neuromuscular efficiency and have a reduced capacity to recover from HL exercises compared to highly trained athletes (*Chiu et al., 2004*; *Khamoui et al., 2009*; *Scott & Docherty, 2004*). Furthermore, HL protocols often require heavy equipment, which can pose practical challenges in certain training environments. As such, HL training may not always be the most effective or feasible method for inducing PAPE, especially for those with limited training experience. Given these considerations, blood-flow restriction (BFR) training has gained significant attention in the scientific community within this context.

Blood-flow restriction (BFR) training involves applying a tourniquet or inflatable cuff to the limbs to partially obstruct venous return or reduce arterial inflow. This technique uses light loads, typically 20–40% of the one repetition maximum (1RM), which contrasts with traditional high-load resistance training (over 70% 1RM). The partial restriction

of blood flow triggers metabolic stress and hypoxia, leading to adaptations similar to those achieved with high-load training, including increased motor unit recruitment and muscle hypertrophy (*De Queiros et al., 2024*; *Loenneke et al., 2012b*; *Sato, 2005*). Empirical studies have substantiated the efficacy of low-load resistance exercises in conjunction with BFR in sport science. For instance, *Doma et al. (2020)* showed that BFR during lunge exercises improved vertical jump height and power output. Similarly, *Wilk et al. (2022)* reported significant increases in power and bar velocity during bench press exercises with BFR. Furthermore, *Miller et al. (2018)* found that BFR, when combined with whole-body vibration or maximum voluntary contraction, enhanced jump performance (*Miller et al., 2018*). These findings suggest that BFR may augment PAPE through mechanisms such as metabolic stress and increased motor unit recruitment. While there is considerable research on PAPE of combining low-load resistance training with BFR, the back squat of a single session of BFR combined with low-load training on specific performance outcomes, such as vertical jump height (VJH) and force production, have not been fully explored.

This study aims to fill this gap by investigating the acute PAPE effects of BFR+low-load (LL) on explosive lower limb performance. Furthermore, by comparing peak and mean countermovement jump (CMJ) metrics, this study provides distinct insights into neuromuscular response, offering a methodological distinction from previous research that has typically focused on either peak performance or long-term outcomes.

## MATERIALS & METHODS

### Subjects

A repeated measures analysis of variance (ANOVA) was conducted using G*Power software (Version 3.1.9.3) to calculate the required sample size. A significance level ($\alpha$) of 0.05 and a statistical power (1-$\beta$) of 0.8 were selected based on conventional standards in exercise science research, which aim to balance the risk of Type I errors (false positives) and Type II errors (false negatives). This ensures a robust detection of significant effects, if present. The power analysis indicated a required sample size of 12 participants. To account for potential dropouts and ensure sufficient power, we recruited 15 trained male athletes for the study. Fifteen trained male participants with diverse athletic backgrounds (American football, basketball, rugby, and track and field) volunteered for this study, the participant characteristics are presented in Table 1. They reported engaging in resistance training at least three times per week for the past two years and confirmed being free of musculoskeletal injuries for at least six months before participation (*Wilk et al., 2019*). After an initial screening, participants were provided with information regarding the potential risks, experimental procedures, and purpose of the study. All participants provided written informed consent. The research protocol was designed in compliance with the ethical guidelines for human experimentation stated in the Declaration of Helsinki and received approval from the Wenzhou Medical University Review Board (2024-002).

### Study design

This study utilized a randomized crossover design, incorporating a familiarization session followed by three experimental sessions in the laboratory, each separated by 72-hour

**Table 1  Participant characteristics.**

| Variable | Mean ± SD |
| --- | ---: |
| Age (years) | 21.7 ± 1.4 |
| Height (cm) | 179.1 ± 5.7 |
| Weight (kg) | 77.2 ± 6.3 |
| BMI (kg/m$^2$) | 23.3 ± 2.3 |
| Thigh circumference L (cm) | 58.5 ± 2.9 |
| Thigh circumference R (cm) | 58.8 ± 3.1 |
| 1RM Back Squat (kg) | 147.3 ± 20.8 |
| 30%1RM Back Squat (kg) | 44.2 ± 6.2 |

**Notes.**

BMI, Body Mass Index; 1RM, 1 repetition maximum.

intervals. The familiarization session involved anthropometric assessments, a standardized warm-up routine, determination of one-repetition maximum (1RM) for the back squat, and a CMJ test. The warm-up comprised a 10-minute sequence: 5 min on a cycling ergometer (Power Max VIII; Konami Corporation, Tokyo, Japan) and 5 min of dynamic stretches. Participants were randomly assigned to one of the three experimental conditions using a computer-generated random number sequence to ensure unbiased allocation. The outcome assessors were blinded to the group assignments to reduce bias. The conditions included: (1) low-load resistance training at 30% of the 1RM for the back squat, combined with blood flow restriction at estimated 80% of individual arterial occlusion pressure (BFR+LL); (2) low-load resistance training without blood flow restriction (LL); and (3) control condition involving passive rest (CON), with no restriction, where participants maintained a static standing position for a total of 360 s, divided into four sets of 45 s with three 60-second intervals. During the experimental sessions, the pneumatic cuffs remained inflated throughout the entire set of exercises to maintain blood flow restriction and were deflated during the one-minute rest periods between sets. Three CMJs were performed at each assessment time point: before (baseline), immediately after (CMJ-0), and at 3 (CMJ-3), 6 (CMJ-6), 9 (CMJ-9), and 12 (CMJ-12) minutes following each condition. All participants completed the prescribed repetition scheme of four sets of 15 repetitions for each trial, and no participants failed to complete the full set and repetition scheme, ensuring consistency across the trials. Participants were instructed to abstain from consuming alcohol and caffeine, as well as to avoid engaging in strenuous physical activity for 24 h prior to each testing session. They were also required to document their food intake in a 24-hour food diary before the initial test and replicate this diet for all subsequent trials. During the testing period, participants were asked to maintain their normal diet and refrain from taking anti-inflammatory drugs and nutritional supplements (*e.g.*, amino acids, creatine, and exogenous antioxidants). These precautions were implemented to avoid potential confounding effects on muscle recovery and performance. Compliance with these controls was monitored through participant self-report and random checks on food diaries before each trial. Any deviations from the prescribed dietary protocols resulted in exclusion from the analysis for that specific trial, and non-compliant data were omitted from the final

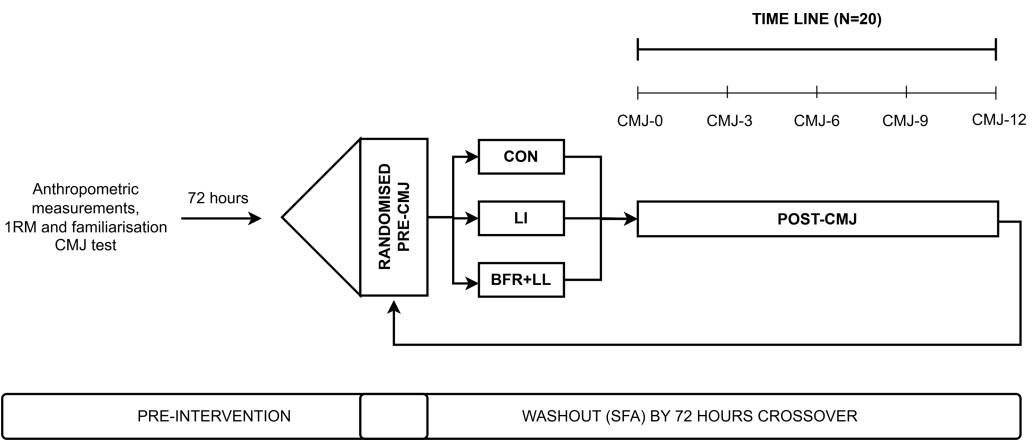

Figure 1  Experimental trials.

dataset. Anti-inflammatory drugs may alter the inflammatory response, which is crucial for muscle repair and adaptation, while nutritional supplements could influence exercise performance or accelerate muscle recovery, thus affecting the study outcomes.

## Procedures

The experimental sessions are depicted in Fig. 1. Upon arriving at the laboratory, the previously described warm-up from the familiarization session was used on each trial. After 5 min of rest, baseline testing for CMJ were performed before the experimental sessions. Subsequently, one of the three trails were performed: (A) low-load resistance exercise at 30% 1RM back squat combined with BFR for four sets of 15 repetitions (BFR+LL): the pneumatic cuffs were inflated to estimated 80% of the participant's individual arterial occlusion pressure (AOP) during each exercise set and deflated during the inter-set rest periods. (B) Low-load resistance exercise without BFR for four sets of 15 repetitions (LL): this condition followed the same exercise protocol but without the application of BFR. (C) Control condition involving passive rest (CON): no exercise and BFR was applied. Participants maintained a static standing position for a total of 360 s, divided into four sets of 45 s with three 60-second intervals. All squat repetitions were executed with a 2-second eccentric phase and a 1-second concentric phase. Between each set, a one-minute rest period was observed, during which subjects were instructed to remain standing without any movement. After each trial, participant completed three CMJ attempt at each assessment timepoint. The peak and average CMJ performance was recorded for further analysis.

## Familiarization session

The initial familiarization session involved anthropometric assessments, standardized warm-up routine, countermovement jump (CMJ), and one-repetition maximum (1RM) back squat test. Once participants arrived at the laboratory, the anthropometric assessments of height and body mass were first recorded. Thigh circumference measurements for both the left and right legs were taken at one-third the distance from the inguinal crease to the

patellar apex. A standardized 10-minute warm-up routine was then conducted, consisting of ergometer cycling (Power Max VIII, Konami Corporation, Tokyo, Japan) for 5 min, followed by dynamic stretching exercises. Subsequently, participants were instructed on the countermovement jump (CMJ) technique and performed six maximal effort jumps. The session concluded with the one-repetition maximum (1RM) back squat test.

### Blood-flow restriction

BFR was performed using an automatic inflated pneumatic system (KAATSU Master Mini, Sato Sports Plaza, Tokyo, Japan). Participants were instructed to sit on a bench with their legs relaxed and extended, ensuring that the knee joint was in a slightly flexed position (approximately 20–30 degrees) to prevent any excessive muscle tension. The distance from the inguinal crease to the superior border of the patella was measured using a tape measure, and a mark was made on the leg at 33% of the total length distal to the inguinal crease (*Loenneke et al., 2012a*). A pneumatic cuff with dimensions of $13 \times 43$ cm was positioned at this marked location to apply BFR pressure accurately. Participants wore pressurized cuffs inflated to 80% of their estimated arterial occlusion pressure (AOP), which was calculated based on their thigh circumference (*Loenneke et al., 2015*; *Murray et al., 2021*).

### One-repetition maximum test (1RM)

The one-repetition maximum (1RM) testing protocol was adhered to according to the guidelines established by the National Strength and Conditioning Association (NSCA), utilizing a smith machine for consistency and safety (*Lawler & Coach, 2013*). Participants were directed to adopt a high bar position for the squat, positioning the barbell above the posterior deltoids. The stance required was slightly wider than shoulder-width with toes pointing outward to initiate the descent. A metronome was utilized to regulate the squatting speed, ensuring a consistent pace of 2 s for the eccentric phase and 1 s for the concentric phase, explicitly avoiding any form of jumping. To maximize safety during the test, at least two experienced trainers were present at all times to supervise and assist, positioned on each side of the barbell to closely observe and guide the movement through both the descent and ascent phases. The determination of all back squat 1RMs was achieved within five attempts, with participants being afforded a 3-minute rest interval between each attempt to ensure optimal performance and recovery.

### Countermovement jump test (CMJ)

For CMJ performance, participants were instructed to place their hands on their waist and were encouraged to jump as high as possible. Beginning from a static upright position, they were told to execute a downward countermovement to a depth of their choosing, immediately followed by an explosive upward leap. It was emphasized that hands should remain on the hips throughout the entire jump to ensure consistency in measuring jump performance. A dual-force platform (model 9260AA, Kistler Instruments Corp, Switzerland), operating at a sampling rate of 1,000 Hz, was employed to capture bilateral three-dimensional kinetic data during the jumps. Vertical jump height (VJH) was calculated using the flight time from the initiation of the jump to landing. Relative peak power output (RPP) was calculated by dividing the peak power output during the jump
by the participant's body weight. Peak power output was obtained from the force-time curve generated by the force platform during the jump, using the formula $Power = Force \times Velocity$, where force is measured in newtons (N) and velocity in meters per second (m/s). Force impulse (FI) was quantified as the integral of the force-time curve from the onset of the jump until takeoff, representing the total force exerted over the duration of the jump, with units in Newton-seconds (Ns). Rate of force development (RFD) was assessed from the slope of the force-time curve during the concentric phase of the jump, calculated as $RFD = \Delta Force / \Delta Time$. The RFD was calculated as the slope between the point where the vertical ground reaction force first exceeded the baseline value by 5% of the participant's body weight and the peak force. All mechanical jump parameters were computed using the MARS software (version 4.0; Kistler Instruments Corp, Switzerland).

## Statistical analyses

Prior to conducting the repeated-measures ANOVA, normality of the data was assessed using the Shapiro–Wilk test ($p > 0.05$). The assumption of sphericity was tested using Mauchly's test, and when violated, the Greenhouse-Geisser correction was applied. A two-way repeated-measures ANOVA was used to analyze the variations in absolute values and percentage changes of CMJ metrics across different protocols (BFR+LL, LL, and CON) and time points (Baseline, CMJ-0, CMJ-3, CMJ-6, CMJ-9, CMJ-12). Bonferroni corrections for multiple comparisons were applied. Results are presented as means ± standard deviations (SD), with statistical significance set at $p < 0.05$. Changes in variables were expressed as percentages relative to baseline. Cohen's d effect sizes (ES) were calculated to quantify the magnitude of observed differences. A medium effect size ($d = 0.5-0.8$) indicates a moderate practical difference in jump performance, which may be noticeable in real-world athletic performance. A large effect size ($d > 0.8$) represents a substantial improvement in performance, suggesting a more meaningful enhancement that could have practical implications for sports performance, particularly in explosive movements such as vertical jumps (*Cohen, 1988*). Statistical analyses were performed using the Prism software (version 9.3.0, GraphPad).

## RESULTS

Statistical analyses showed significant condition × time interactions for VJH, RPP, FI, and RFD with detailed results provided in Table 2. Mean and peak absolute values for vertical jump parameters are shown in Tables 3 and 4, respectively, while percentage changes are illustrated in Fig. 2.

The two-way ANOVA revealed significant condition × time interactions for VJH (F (4.535, 63.50) = 16.94, $p < 0.001$), RPP (F(10, 140) = 11.76, $p < 0.001$) and FI (F10, 140 = 11.00, $p < 0.001$) (Table 2). Multiple comparisons analysis revealed that BFR+LL significantly ($p < 0.05$) improved VJH (46.93 ± 3.78 cm, ES = 0.76), RPP (61.13 ± 4.31 w/kg, ES = 0.42), and FI (235.0 ± 25.3 Ns, ES = 0.33) compared with CON (43.79 ± 4.13 cm, 59.78 ± 4.45 w/kg, 227.7 ± 25.4, Ns) and LL (44.37 ± 4.32 cm, 59.16 ± 4.66 w/kg, 228.0 ± 28.7 Ns) at CMJ-12 (Table 3). The percentage changes from baseline were 6.3 ± 2.2%, 3.0 ± 2.4%, and 3.9 ± 2.9%, respectively. However, there was

**Table 2  ANOVA results for main and interaction effects on VJH, RPP, FI, and RFD.**

|  | Condition × Time | | Time | | Condition | |
|---|---|---|---|---|---|---|
|  | F | P | F | P | F | P |
| VJH (cm) | 16.94 | $P < 0.001$ | 25.54 | $P < 0.001$ | 0.57 | $P = 0.559$ |
| RPP (w/kg) | 11.76 | $P < 0.0001$ | 20.12 | $P < 0.0001$ | 0.50 | $P = 0.6106$ |
| FI (ns) | 11.00 | $P < 0.0001$ | 23.69 | $P < 0.0001$ | 0.20 | $P = 0.8237$ |
| RFD | 1.46 | $P = 0.1571$ | 1.256 | $P = 0.2927$ | 0.98 | $P = 0.0867$ |

Notes.

Abbreviations: VJH, Vertical Jump Height; RPP, Relative Peak Power Output; FI, Force Impulse; RFD, Rate of Force Development.

The significance level for all comparisons is $p < 0.05$

**Table 3  Absolute mean values of CMJ parameters post three conditioning sessions.** Values are means ± SD ($n = 15$).

|  | Condition | Baseline | T0 | T12 | F | P | Partial eta |
|---|---|---|---|---|---|---|---|
| VJH (cm) | CON | 44.99 ± 4.37 | 44.24 ± 4.61 | 43.79 ± 4.13 | | | |
| | LL | 45.39 ± 3.98 | 39.90 ± 4.88[*#] | 44.37 ± 4.32 | 37.416 | <0.001 | 0.471 |
| | BFR+LL | 44.17 ± 3.60 | 40.15 ± 3.87[*#$] | 46.93 ± 3.78[*#] | | | |
| RPP (w/kg) | CON | 59.70 ± 4.20 | 59.47 ± 4.63 | 59.78 ± 4.45 | | | |
| | LL | 60.25 ± 4.71 | 56.74 ± 5.91[*#] | 59.16 ± 4.66 | 23.874 | <0.001 | 0.362 |
| | BFR+LL | 59.36 ± 4.26 | 54.40 ± 4.55[*#$] | 61.13 ± 4.31[*#$] | | | |
| FI (ns) | CON | 228.0 ± 28.5 | 226.9 ± 28.7 | 227.7 ± 25.4 | | | |
| | LL | 228.5 ± 25.7 | 221.1 ± 28.8[*#] | 228.0 ± 28.7 | 27.306 | <0.001 | 0.394 |
| | BFR+LL | 226.4 ± 25.8 | 215.0 ± 23.2[*#$] | 235.0 ± 25.3[*#$] | | | |
| RFD (n/s) | CON | 10,327 ± 2,735 | 10,788 ± 2,770 | 10,662 ± 2,926 | | | |
| | LL | 9917 ± 2844 | 9418 ± 2330 | 9868 ± 3967 | 1.552 | $P = 0.205$ | 0.036 |
| | BFR+LL | 10508 ± 3902 | 9813 ± 3463 | 11480 ± 3991 | | | |

Notes.

CON, conditioning with passive rest; LL, low-load resistance exercise using 30% 1RM back squat without blood flow restriction; BFR+LL, low-load resistance exercise with blood flow restriction; CMJ, Countermovement jump; VJH: Vertical jump height; RPP, Relative peak power output; FI, Force impulse; RFD, Rate of force development.

[*]$P < 0.05$, compared with the baseline in the same group.

[#]$P < 0.05$, compared with CON at the same time point.

[$]$P < 0.05$, compared with LL at the same time point

a significant decline in performance ($p$ <0.05) observed immediately after BFR+LL, with VJH (40.15 ± 3.87 cm, ES = −0.89), RPP (54.40 ± 4.55 w/kg, ES = −1.11), and FI (215.0 ±23.2 Ns, ES = −0.51) compared to CON (44.24 ± 4.61 cm, 59.47 ± 4.63 w/kg, 226.9 ± 28.7, Ns) and LL (39.90 ±4.88 cm, 56.74 ± 5.91 w/kg, 221.1 ± 28.8 Ns) at CMJ-0. The percentage changes from baseline were −9.1 ± 5.1%, −8.3 ± 4.3%, and 5.0 ± 2.9%, respectively.

There were no significant differences in RFD at any time points after the three protocols. Moreover, the peak CMJ performance was not different between baseline and either after CON, LL or BFR+LL (Table 4).

## DISCUSSION

This study aimed to investigate the effect of low-load resistance training combined with bilateral blood flow restriction on inducing post-activation performance enhancement

**Table 4  Absolute mean values of CMJ parameters post three conditioning sessions.**

| Conditions | Variables | Baseline | CMJ-0 | CMJ-3 | CMJ-6 | CMJ-9 | CMJ-12 |
|---|---|---|---|---|---|---|---|
| | | | | **Time points** | | | |
| CON | VJH (cm) | 44.99 ± 4.37 | 44.24 ± 4.61 | 44.12 ± 4.04 | 45.03 ± 4.48 | 44.48 ± 3.71 | 43.79 ± 4.13 |
| | RPP (W/kg) | 59.70 ± 4.20 | 59.47 ± 4.63 | 59.08 ± 4.64 | 59.63 ± 4.98 | 59.15 ± 4.52 | 59.78 ± 4.45 |
| | FI (Ns) | 228.0 ± 28.5 | 226.9 ± 28.7 | 226.3 ± 27.5 | 227.4 ± 27.8 | 227.7 ± 26.0 | 227.7 ± 25.4 |
| | RFD (N/s) | 10,327 ± 2,735 | 10,788 ± 2,770 | 10,677 ± 3,341 | 10,160 ± 2,943 | 10,147 ± 2,906 | 10,662 ± 2,926 |
| LL | VJH (cm) | 45.39 ± 3.98 | 39.90 ± 4.88[*#] | 44.18 ± 4.79 | 44.47 ± 4.38 | 44.34 ± 4.16 | 44.37 ± 4.32 |
| | RPP (W/kg) | 60.25 ± 4.71 | 56.74 ± 5.91[*#] | 59.12 ± 5.76 | 59.42 ± 5.07 | 59.53 ± 4.44 | 59.16 ± 4.66 |
| | FI (Ns) | 228.5 ± 25.7 | 221.1 ± 28.8[*#] | 226.2 ± 26.6 | 229.2 ± 27.6 | 228.1 ± 27.6 | 228.0 ± 28.7 |
| | RFD (N/s) | 9917 ± 2844 | 9418 ± 2330 | 9101 ± 1887 | 9994 ± 3256 | 10043 ± 3893 | 9868 ± 3967 |
| BFR+LL | VJH (cm) | 44.17 ± 3.60 | 40.15 ± 3.87[*#$] | 42.59 ± 3.83 | 43.85 ± 3.99 | 44.26 ± 3.79 | 46.93 ± 3.78[*#$] |
| | RPP (W/kg) | 59.36 ± 4.26 | 54.40 ± 4.55[*#$] | 57.71 ± 4.29 | 59.03 ± 4.38 | 59.08 ± 4.47 | 61.13 ± 4.31[*#$] |
| | FI (Ns) | 226.4 ± 25.8 | 215.0 ± 23.2[*#$] | 222.2 ± 25.4 | 225.6 ± 25.7 | 229.8 ± 26.3 | 235.0 ± 25.3[*#$] |
| | RFD (N/s) | 10,508 ± 3,902 | 9,813 ± 3,463 | 9,755 ± 2,768 | 10,870 ± 3,289 | 10,957 ± 3,457 | 11,480 ± 3,991 |

**Notes.**

Values are means ± SD ($n = 15$).

CON, conditioning with passive rest; LL, low-load resistance exercise using 30% 1RM back squat without blood flow restriction; BFR+LL, low-load resistance exercise with blood flow restriction; CMJ, Countermovement jump; VJH, Vertical jump height; RPP, Relative peak power output; FI, Force impulse; RFD, Rate of force development.

[*] $P < 0.05$, compared with the baseline in the same group.

[#] $P < 0.05$, compared with CON at the same time point.

[$] $P < 0.05$, compared with LL at the same time point

(PAPE) in vertical jump performance. The key findings of this study revealed that BFR+LL resulted in improved vertical jump height (VJH), relative peak power output (RPP), and force impulse (FI) at the 12th minute after the completion of the protocol, despite a temporary decrease in these measures immediately after the protocol. Additionally, the study found no significant changes in the rate of force development (RFD) and no notable differences in peak countermovement jump (CMJ) performance across the three tested conditions.

Previous studies have shown that BFR combined with resistance exercise can enhance performance within 12 min post-exercise, depending on the training protocol and intensity. For example, *Doma et al. (2020)* observed improvements in jump height and power output with BFR during lunge exercises, while *Wilk et al. (2022)* reported increased power and bar velocity in bench press exercises (*Doma et al., 2020*; *Wilk et al., 2022*). *Miller et al. (2018)* also found positive effects of BFR combined with whole-body vibration or maximum voluntary contraction on jump performance (*Miller et al., 2018*). The mechanisms underlying post-activation performance enhancement (PAPE) following BFR+LL training remain unclear. While metabolites such as lactate and reactive oxygen species (ROS) are commonly proposed to play a role in potentiating performance, their precise contributions to the acute performance enhancement observed in this study are speculative. Lactate accumulation during BFR+LL training likely contributes to metabolic stress, which has been associated with short-term improvements in performance. However, the exact mechanisms by which lactate may enhance performance are still not fully understood. Similarly, while

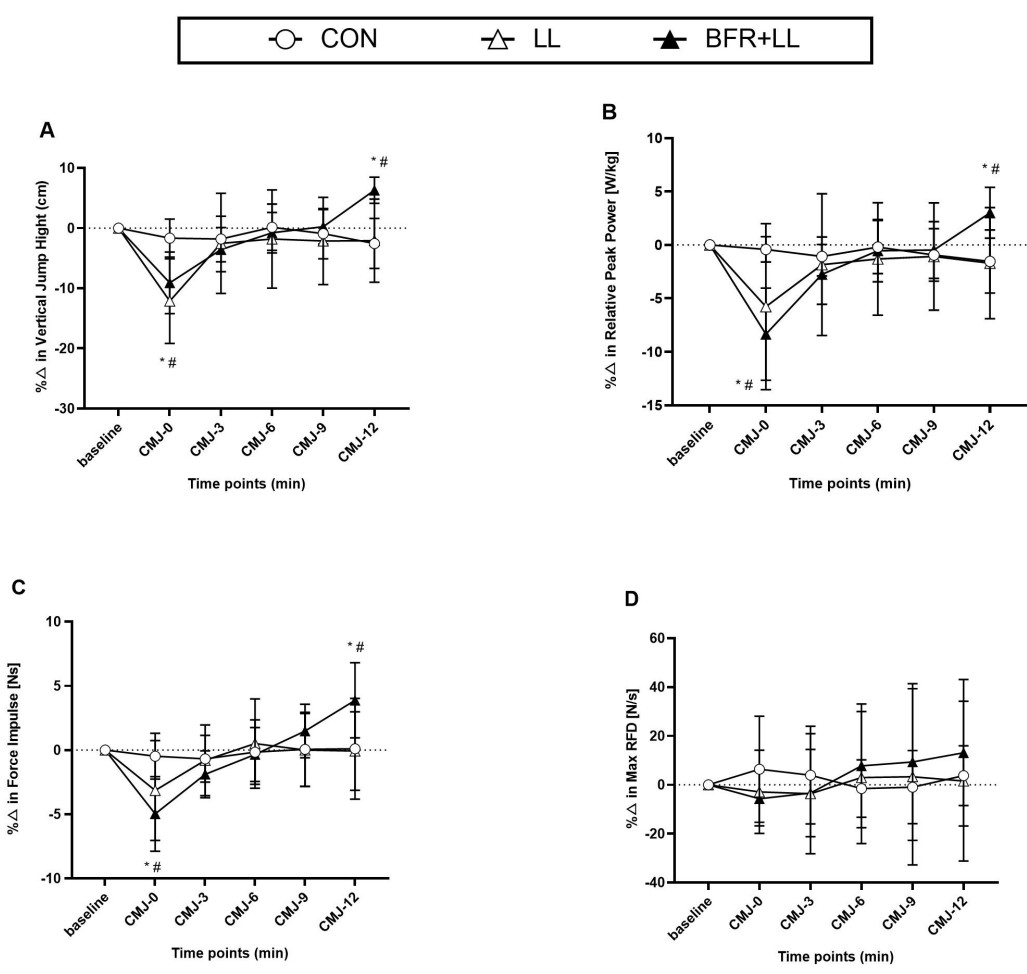

**Figure 2** **Percentage changes in (A) VJH, (B) RPP, (C) FI, and (D) RFD across three training conditions relative to baseline.** CON, conditioning with passive rest; LL, low-load resistance exercise using 30% 1RM back squat without blood flow restriction; BFR+LL, low-load resistance exercise with blood flow restriction; CMJ, Countermovement jump; VJH, Vertical jump height; RPP, Relative peak power output; FI, Force impulse; RFD, Rate of force development. * $P < 0.05$, significantly different from baseline for BFR+LL. # $P < 0.05$, significantly different from baseline for LL.

ROS are traditionally linked to long-term adaptations like muscle hypertrophy, they may also contribute to acute performance enhancement in BFR training through specific signaling pathways that support recovery and performance. The role of these metabolites in potentiating performance remains uncertain, and future research is needed to clarify these mechanisms. These mechanisms contribute to the acute performance enhancements observed post-BFR+LL training. While our study's methodology did not directly allow us to verify the contributions of lactate and ROS to the PAPE effects, our findings suggest that the current unrestricted low-load resistance exercise protocol may not effectively improve CMJ performance. This could be due to the nature of the repetitions, which may not provide the necessary explosive load required to enhance vertical jump performance. Additionally, the current study employed a protocol of four sets of 15 repetitions, which induced a

significant PAPE response in vertical jump performance. The observed effects may be linked to the alignment of the kinematic characteristics of the conditioning contractions with the performance measure (*i.e.,* vertical jump), as studies suggest that the induction of PAPE is more effective when these characteristics are similar (*Doma et al., 2019*; *Doma et al., 2016*). However, as an inducing protocol, the use of multiple sets of 15 repetitions in this study was unavoidable for generating sufficient metabolic stress and muscle fiber recruitment. Future studies could explore alternative exercise schemes, such as varying repetition tempos or using higher-intensity protocols, to further optimize the induction of PAPE and maximize its effects on explosive performance. Consequently, by correlating these metrics with post-activation performance enhancement (PAPE), our study provides empirical support for the efficacy of BFR+LL training in eliciting PAPE. This offers valuable insights for athletes and coaches aiming to optimize training outcomes for explosive sports performance.

It is interesting to note that CMJ performance decreased following BFR+LL at CMJ-0 in the current study. This finding is consistent with previous studies that have reported a temporary decrease in performance after similar interventions. For example, *Fowles & Green (2003)* noted a diminution in muscle strength after a maximal voluntary contraction of the quadriceps muscle, highlighting the immediate impact of such exercises on muscle function. *Teixeira et al. (2018)* reported an increase in metabolic stress identified with blood lactate measurements and a decrease in muscle activation identified with electromyography (EMG) measurements after high-intensity resistance exercise. Similarly, *Scott & Docherty (2004)* observed a temporary decrease in vertical and horizontal jump performances in nineteen resistance-trained subjects after submaximal intensity (5 RM) deep squat exercises. The observed decrease in CMJ performance immediately after the BFR+LL protocol can be attributed to acute fatigue mechanisms. Central nervous system (CNS) fatigue, potentially associated with the accumulation of neurotransmitters such as serotonin and glutamate, along with peripheral factors, including disruptions in cross-bridge cycling, may have contributed to the temporary performance decrement. This mechanism aligns with findings from prior studies reporting neurotransmitter imbalance and altered muscle contraction kinetics post-exercise (*Meeusen et al., 2006*; *Myburgh, 2004*; *Tornero-Aguilera et al., 2022*). Additionally, peripheral nervous system (PNS) fatigue, which refers to fatigue originating from the peripheral nervous system, may involve disruptions in muscle contraction and cross-bridge cycling. The consistent application of BFR during each set of the low-load resistance exercise protocol might exacerbate this accumulation, further inducing PNS fatigue (*Myburgh, 2004*; *Tornero-Aguilera et al., 2022*). These fatigue mechanisms are consistent with prior studies that have reported short-term reductions in muscle performance following high-intensity exercises (*Macdougall & Sale, 2014*; *Tornero-Aguilera et al., 2022*). Hence, when administering PAPE protocols like BFR+LL, sufficient recovery time must be provided to mitigate these fatigue effects and optimize performance gains. This suggests that while BFR+LL can enhance muscle performance through PAPE in the post-exercise period, it may initially lead to a decrease in performance due to the acute onset of fatigue-related mechanisms (*Tornero-Aguilera et al., 2022*).

In the present study, RFD did not show any significant differences ($p > 0.05$), which contrasts with previous studies (*Doma et al., 2020*; *Sun & Yang, 2023*; *Zheng et al., 2022*). While it has been suggested that RFD may not always be a strong predictor of jump performance, there are several potential reasons for the lack of significant changes in our study. First, the RFD measurement protocol used may not have been sensitive enough to detect subtle changes in force development, particularly considering the low-load nature of the BFR+LL protocol, which may not provide sufficient neural responses necessary for a significant increase in RFD, as typically seen in higher-load resistance training. Previous studies have shown that RFD measurements can be influenced by factors such as task specificity and participant characteristics. Second, the BFR+LL protocol in our study, which combined low-load squats with blood flow restriction, might have induced different neural responses compared to the protocols used in prior studies. For instance, *Doma et al. (2020)* used lunge exercises with BFR, which likely engaged different muscle groups and movement patterns, potentially leading to different neural responses (*Doma et al., 2020*). These considerations help explain the lack of significant changes in RFD in our study. Recent research, including *Miller et al. (2023)*, has indicated that the relationship between RFD and VJ height is not as strong as previously thought, suggesting other factors might be more critical in determining vertical jump performance (*Miller et al., 2023*). Nonetheless, the observed mean RFD values align with findings from several studies which have indicated a weak correlation between RFD and vertical jump outcomes (*Haff et al., 2000*; *Marcora & Miller, 2000*; *Wilson et al., 1995*; *Young & Bilby, 1993*). For instance, *Miller et al. (2023)* demonstrated that the relationship between RFD and VJH is not as strong as previously thought, suggesting that other neuromuscular factors might play a more critical role in determining vertical jump performance. *Wilson et al. (1995)* reported the absence of a significant link between RFD and countermovement jump (CMJ) performance, advocating concentric tests as more reliable for assessing RFD. Similarly, *Haff et al. (2000)* found no correlation between RFD and either CMJ or squat jump (SJ) performance among collegiate athletes. Such variations in findings could stem from methodological differences, participant gender (*Marcora & Miller, 2000*), or training status (*Young & Bilby, 1993*) across studies. These findings align with our observation that RFD did not show significant differences across the conditions tested in our study. Given these disparities and the consistent reports of poor correlations between RFD and vertical jump performance, it becomes imperative to approach RFD as a performance metric for vertical jumps with caution, especially in the context of male collegiate athletes. The apparent inconsistency in its reliability underscores the need for careful consideration of the factors that may influence the relationship between RFD and jump performance, suggesting that RFD may not always serve as a direct or reliable indicator of vertical jump capability (*McLellan, Lovell & Gass, 2011*).

Furthermore, the analysis of peak countermovement jump (CMJ) performance did not reveal any differences, echoing a broader dialogue within recent research about whether to prioritize peak or mean CMJ performance for post-intervention evaluation (*Al Haddad, Simpson & Buchheit, 2015*; *Hetherington, 1973*). Despite the varied applications of CMJ tests in recent studies, our results showed no significant distinctions between peak (effect

size (ES) = 0.32, 95% confidence interval (CI) [0.05–0.65]) and mean (ES = 0.35, 95% CI [0.02–0.62]) CMJ height, indicating that both peak and mean measurements could be effectively used to monitor changes in CMJ performance. However, a meta-analysis by *Claudino et al. (2017)* pointed out that averaging CMJ parameters could potentially offer a more sensitive approach than focusing solely on peak values for tracking vertical jump performance, especially when increasing the sample size for such analyses. Given these insights, and to mitigate measurement inaccuracies while improving the precision of data, our findings advocate for the adoption of mean values as more reliable indicators for assessing neuromuscular status. Mean values offer a more stable assessment of VJ performance by mitigating the influence of outliers and single exceptional performances, ensuring a reliable indicator of consistent performance across multiple trials.

In summarizing our findings, it is crucial to acknowledge several limitations within our study. First, the investigation focused exclusively on the impact of blood-flow restriction combined with low-load (BFR+LL) resistance training solely on vertical jump performance. This narrow focus limits the applicability of our results to other sport-specific performance metrics, such as sprint speed and change-of-direction capabilities. Second, a significant limitation is the absence of a heavy load (HL) resistance training group, which precludes direct comparison between these two training modalities. While BFR+LL has shown promise as a practical alternative, particularly in contexts where high-load training is not feasible, the superiority or equivalence of BFR+LL to HL training cannot be concluded from our results alone. Future research should include HL training groups to provide a comprehensive evaluation of BFR+LL training's relative effectiveness and to better understand the underlying physiological mechanisms. Another key limitation lies in the use of percentage changes in CMJ performance. While these changes provide a practical measure of performance enhancement, they may not fully capture the nuanced relationship between muscle strength and jump height. *Chatlaong et al. (2024)* highlighted the interpretive and statistical challenges associated with ratios or percentages when analyzing muscle strength relative to muscle size, which could also be relevant here (*Chatlaong et al., 2024*). Percent change may not be the most precise way to evaluate improvements in muscle performance, especially in strength and explosive power assessments. To better capture these effects, future studies could incorporate alternative metrics, such as absolute strength measurements or muscle cross-sectional area, which would offer a more comprehensive understanding of the relationship between muscle adaptations and performance outcomes. Lastly, the study's generalizability is constrained by its small sample size and lack of gender diversity. Future research with larger, more gender-balanced cohorts is essential to elucidate potential sex-specific responses to PAPE following BFR+LL, thereby enhancing our comprehension of its effects across a broader athletic population.

## CONCLUSIONS

We acknowledge that both blood-flow restriction (BFR) training and heavy load lifting have their own practical challenges and advantages. BFR training can be particularly useful

in scenarios with limited access to heavy equipment or in specific rehabilitation contexts. Conversely, low volumes of heavy but non-fatiguing loads have been shown to be highly effective and are extensively used in both research and practical applications. Therefore, the choice between BFR and heavy load training should be guided by the specific needs and circumstances of the athlete. In conclusion, this finding underscores the potential of this training methodology as a viable strategy for enhancing explosive lower limb performance.

## ACKNOWLEDGEMENTS

We wish to thank all of the participants in the present study. Additionally, we acknowledge the laboratory members for their technical support.

### Funding
The authors received no funding for this work.

### Competing Interests
The authors declare there are no competing interests.

### Author Contributions
- Zhanfei Zheng conceived and designed the experiments, performed the experiments, prepared figures and/or tables, authored or reviewed drafts of the article, and approved the final draft.
- Zhipeng Wang analyzed the data, authored or reviewed drafts of the article, and approved the final draft.
- Changyuan Duan performed the experiments, prepared figures and/or tables, and approved the final draft.
- Yu Zhang conceived and designed the experiments, analyzed the data, prepared figures and/or tables, authored or reviewed drafts of the article, and approved the final draft.
- Xiaowei Wang conceived and designed the experiments, performed the experiments, analyzed the data, authored or reviewed drafts of the article, and approved the final draft.
- Xixi Miao performed the experiments, analyzed the data, prepared figures and/or tables, and approved the final draft.
- Haiping Liu conceived and designed the experiments, performed the experiments, prepared figures and/or tables, and approved the final draft.

### Human Ethics
The following information was supplied relating to ethical approvals (i.e., approving body and any reference numbers):

Wenzhou Medical Universitygranted Ethical approval to carry out the study within its facilities (Ethical Application Ref:2024-002).

## Data Availability

The raw data is available in the Supplementary Files.

## Supplemental Information

Supplemental information for this article can be found online at http://dx.doi.org/10.7717/peerj.19272#supplemental-information.

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
