# Peer review of "Effects of bilateral low-load resistance training with blood-flow restriction on post-activation performance enhancement in male collegiate athletes"

_PeerJ, doi:10.7717/peerj.19272_

## Round 0.1 · original submission · Major Revisions

Authors found your paper of interest but there are significant revisions required. Reviewers provided comments on the literature used to support certain claims, the reporting of results, and discussion of your findings in the broader context of the literature. Please make sure you address each of the reviewers concerns.

Reviewer 1 ·

Basic reporting

Abstract:

Line 20: "experienced male strength trainers," is somewhat vague. Clarifying their training background (e.g., "male strength trainers with at least two years of experience in resistance training") would add precision if space is permitted.
Please also consider adding in the age range or mean if relevant to interpreting results.
Line 33: The term "peak CMJ" is used in the results but is not introduced in the methods. If CMJ refers to countermovement jump, clarifying this earlier could prevent confusion.
Line 34: "Conlusion" should be corrected to "Conclusion."
The conclusion could be more specific to reflect both the benefits and limitations observed. Instead of a general statement, consider:
• "The study suggests that BFR+LL resistance training may enhance acute vertical jump performance 12 minutes post-exercise, despite an initial decline in performance immediately following the protocol."
Materials and Methods:

Line 93: While the mention of G*Power software for sample size calculation is good, it might be helpful to briefly explain the rationale behind choosing a significance level of 0.05 and a power of 0.8.
Additionally, it was mentioned using G*Power but don’t provide the actual sample size calculated from the analysis. Including the exact sample size required based on the power analysis would make the methodology more transparent.
Line 95: It’s mentioned that participants were "trained" athletes, but the definition of "trained" is not provided. What specific criteria (e.g., years of experience, frequency of resistance training) were used to define "trained"?
Line 135: Blood flow restriction is not typically supposed to occlude the arteries, so the term “no occlusion” should be “no restriction.”
Line 204: It was mentioned that normality was assessed using the Shapiro-Wilk test, but it might be useful to briefly note any additional assumptions checked for the repeated-measures ANOVA, such as sphericity. If these assumptions were not checked or if they were confirmed, it would be useful to mention this.
Line 206: Do keep in mind that the percentage in changes of CMJ may be a limitation. Although it is not strength per se, still a consideration.
Line 363: "change-of-direction" does not need an abbreviation.

Figures and Tables:
Figure 1: Consider not elongating the images as it makes it more difficult to interpret.
Figure 2A: On the y-axis, it looks like jump height is spelled incorrectly.
Table 2 should be revised: Tables would be more beneficial as showing parameters. These F and p-values have been noted in the results section. Pre and post values of the outcomes would be more beneficial.
Table 3: There should be some formatting changes completed. Please use Table 1 as an example.

Experimental design

Study Design:

Lines 124–126: It was stated that participants should refrain from taking anti-inflammatory drugs and nutritional supplements, but it could be useful to specify why this is important. For instance, anti-inflammatory drugs might alter muscle recovery and performance, while nutritional supplements (e.g., creatine, amino acids) may influence exercise performance or muscle recovery.
Materials and Methods:

Line 93: It might be helpful to explain why a significance level of 0.05 and a power of 0.8 were chosen, specifically in exercise science research.
Statistical Considerations:

Line 204: It might be useful to briefly note any additional assumptions checked for the repeated-measures ANOVA, such as sphericity. If these assumptions were not checked or confirmed, it would be useful to mention this to provide a complete picture of the statistical approach.
Additional Comments on Study Design:

Line 212: The interpretation of Cohen's d is clearly outlined, which is excellent. However, providing a brief explanation of the practical significance of medium and large effect sizes in the context of your study would help contextualize the results.
Line 318: RFD did not show significant changes and contrasting the current study’s results with previous studies (e.g., Zheng et al., 2022, Doma et al., 2020). However, it would be helpful to explore why RFD did not show the expected results.

Validity of the findings

Materials and Methods:

Line 95: Defining "trained" participants more clearly will strengthen the internal validity of your study.
Study Design:

Lines 162–164: Acknowledge the limitations of using these equations, including references to Loenneke’s group.

Interpretation of Results:

Line 318: RFD did not show significant changes; it would be helpful to explore why this outcome occurred, possibly due to the sensitivity of the RFD measurement protocol or how the BFR+LL protocol influenced neural factors differently.

Additional comments

Introduction:

Line 41: It would be better for the reader if the term "contractile activity" is not abbreviated and simply worded out as such.
Line 43: Minor grammar and formatting issues (e.g., spacing errors). Additionally, clarify whether "PAP" refers to "PAPE" or "post-activation potentiation" (if different concepts).
Lines 46–51: Streamline repeated concepts to improve clarity. Consider merging sentences that overlap, like the physiological mechanisms of PAPE.
Line 66: Avoid stating that BFR training is known as KAATSU training. KAATSU is a form of BFR, not synonymous with it.
Line 68: Replace hyphen (-) with an en dash (–) for the range (e.g., 20–40%).
Line 72: Minor grammar and formatting issues throughout, such as missing conjunctions and inconsistencies in referencing styles (e.g., "Sato and Y., 2005" is unclear).
Additional Considerations:

Line 228: The term "Ns" is used to refer to force impulse (FI) values, but it is unclear whether this refers to "Newton-seconds" or something else. Clarify or define the abbreviation.
Lines 255–271: Great details of previous work, but the section could be condensed for clarity and conciseness.

Annotated reviews are not available for download in order to protect the identity of reviewers who chose to remain anonymous.

Reviewer 2 ·

Basic reporting

Abstract

Clearly state the aim of the study.
Provide more detailed information about the study participants, including age, relative strength levels, and any other relevant characteristics.
Specify which type of Vertical Jump Height (VHJ) was measured.
In the results section, clarify whether you are referring to the repeated measures ANOVA results or the post-hoc comparisons. It seems you are describing post-hoc results rather than the ANOVA findings, so please make this distinction clear.

Introduction

Line 53: Remove the unnecessary capital letter in "contractile."
Lines 83–84: Add a paragraph discussing the potential for using Blood Flow Restriction (BFR) to induce Post-Activation Performance Enhancement (PAPE). Clearly indicate the gap in the literature that your study addresses and explain how your research fills this gap. In the final paragraph, discuss the possible practical applications of the study’s findings.

Experimental design

Line 93: You refer to the sample size calculation, but please include the specific effect size used and the calculated required sample size for clarity.

Line 128: Provide a more detailed description of the Blood Flow Restriction (BFR) procedure. Was the BFR applied continuously throughout the intervention, or were there inter-set breaks? Please include this information for better clarity.

Line 156: Specify the body position of participants during inflation to ensure clear replication of your protocol.

Line 202: Did you test for data homogeneity and sphericity? If so, mention the specific tests used to check these assumptions.

Validity of the findings

Results

Ensure that each table is cited in the text at the appropriate point where you discuss specific results. Place each table immediately after the paragraph or sentence where it is first referenced to help readers follow the data and findings.

Review the text to make sure that each result discussed is directly supported by data in the tables. Clarify if the values reported are mean values, standard deviations, or any other statistical measure.

Discussion & Conclusion

No changes are required as this section is already well-written.

Additional comments

Please specify the exact type of jump instead of using "vertical jump" ( countermovement jump/CMJ).

Provide a detailed description of the Blood Flow Restriction (BFR) procedure. Include steps taken before the study to establish the required parameters, as well as the specifics of the BFR application during the intervention.

This is a godo, and I hope these comments will be helpful in further improving it.

Reviewer 3 ·

Basic reporting

Abstract:
Line 18 (Background). “The effects of low-load (LL) resistance training with blood-flow restriction (BFR) on Post-Activation Performance Enhancement (PAPE) have yet to be examined.” I am not certain if this statement is true (for example, https://pmc.ncbi.nlm.nih.gov/articles/PMC7312758/).

Line 27 (Results). I believe that what the readers want to know is whether the blood flow restriction augments PAPE over free-flow and control conditions. Representing time effect compared to baseline adds little information. The authors are recommended to rewrite the results section.

Line 34 (Conclusion). The potential benefit was only observed in one time point with one outcome, and I am not totally sold on whether the study supports the efficacy of BFR as enhancing vertical jump performance.

Introduction
Line 40-43. I am not sure how Wisloff et al., Dobbs et al., and Shalfawi et al. are fitting here. Those three studies did not talk about PAPE.

Line 41. The “contractile activity” was abbreviated as CA, but CA was not really used in the manuscript. Could this be taken out?

Line 47. Could you elaborate on what is muscle fiber sensitivity and how it relates to PAPE?

Line 50. Any particular reason why contractile activity is written as “Contractile” instead? I see that throughout the manuscript.

Line 52. “individual’s unique physiological characteristics (Chiu et al., 2003)” This “physiological characteristics” is very vague. Could this be elaborated?

Line 58-59. “… high-load (HL) exercises might cause … with recreational training backgrounds.” What makes individuals with recreational training backgrounds particularly difficult to achieve potentiation with high-load? Substantial muscle fatigue can be experienced by any individual depending on how the sets are carried out.

Line 60-61. “Additionally, implementing such HL … practical challenges.” I am not certain if this is a strong argument that adds any point regarding the use of high-load. I personally think that you don’t necessarily need to bring high-load protocol down to bring the low-load discussion. For example, the authors follow by stating that low-load is more “safer and more practical,” which I am not if it is true or not.
Line 63-64. “… albeit possibly less potent in eliciting PAPE compared to HL.” Do you have any references that support this claim?

Line 64-65. “Given these considerations, blood-flow restriction (BFR) training has gained significant attention in the scientific community within this context.” Doesn’t this contradict with the statement made in the abstract where the authors mentioned that BFR on PAPE has not been examined (which again I don’t think that is the case)?

Line 75-76. Lixandrao et al. is one study that states that muscle strength achieved through BFR training is comparable to those attained with high-load training, but that is not the case when looking other literatures of high vs. low-load BFR on maximal strength.

Line 77-80. These are incomplete sentences. Please rewrite for clarity.

Line 82-83. “… the acute effects of a single session of BFR combined with low-load training on the lower limbs have yet to be fully explored.” This statement is not true. There are numerous acute studies on the lower body with BFR. The authors may need to specify which outcome of interest is discussed here.

Line 85. “Building on previous research” Which specific study are you referring to? Could this be cited?

Line 86. Please change to “hypothesize”.

Experimental design

Methods
Line 130. “the previously described warm-up…” It says previously described, but I don’t think it was not mentioned until the “Familiarization session” section.

Line 162-163. As AOP was estimated, I think it is important to specify that it was estimated 80% AOP throughout the manuscript.

Line 206. The results (including figures) might be presented with absolute values instead of percentage changes (https://pubmed.ncbi.nlm.nih.gov/22760546/)

Validity of the findings

Results
Line 222-226. In the case where you found the interaction, I would leave the main effect out, and get into the multiple comparisons.

For Figure 2, wouldn’t be more beneficial to know whether the protocols changed differently (interaction) instead of describing the difference from baseline (* and #)?

Line 227-231. Did the effect of free flow condition exceeded the control condition?

Line 231-234. “However, there was a … “ Compared to which condition did BFR+LL observe the reduction?

Discussion
Line 245-247. The sentence is incomplete. Please rewrite for clarity.

Line 247-249. “To our knowledge, this study is the first to…

Line 268-271. With regard to mechanism, could you elaborate how lactate and reactive oxygen species affect performance outcome?

Line 274-276. “… our findings suggest that … CMJ performance.” If the effect of free flow condition exceeded the control group (please see comment on result section), can’t it be at least mentioned that exercise itself has PAPE effect?

Line 298-302. With regard to fatigue mechanism, could the authors provide any reference to support the claim?

Line 307-314. I think this might be an overreach. It is nice to mention potential mechanisms but none of those listed factors were measured in the current study. I don’t think listing all those factors repetitively is necessary. Could this be trimmed down?

Additional comments

Thank you for submitting your research to the PLOS ONE. The authors have presented an important topic regarding the blood flow restriction training on PAPE. The authors should be commended for the efforts in conducting the study.

---

## Round 0.2 · Minor Revisions

All reviewers found merit but Reviewer 1 and 3 have a few comments that need to be addressed in the next revised manuscript.

Reviewer 1 ·

Basic reporting

The text uses clear, professional English throughout, with proper citation of relevant literature. The definitions and concepts, such as PAPE, conditioning activity, and blood-flow restriction training, are described concisely and accurately, which helps provide context for a broad audience.

While the text outlines a strong context, it could benefit from a brief summary of how the proposed study differs methodologically or conceptually from prior research.

Experimental design

Dietary and lifestyle controls (e.g., caffeine and alcohol abstinence, food diary replication) are commendable. However, these controls rely on participant compliance. State whether compliance was monitored or how deviations were handled.

Validity of the findings

The "p" for p-values should be in italics.

There is a lack of follow-up on the implications of the non-significant findings (e.g., for RFD and peak CMJ performance). The null findings could be explored further to explain why BFR+LL did not enhance RFD or peak CMJ performance.

Reviewer 2 ·

Basic reporting

Nothing to add.

Experimental design

Nothing to add.

Validity of the findings

Nothing to add.

Additional comments

Nothing to add.

Reviewer 3 ·

Basic reporting

Introduction
Line 64-74. Reply to rebuttal. Thank you for the update on the manuscript. I am still uncertain about the high load protocol being more fatigue inducing than low load protocol, and that relates to training status. There is a sentence in Chiu et al. (2004) describing that trained individuals may tolerate intensive exercise better but this experimental study itself is not a source to demonstrate that. They only recruited recreationally trained individuals, and there were no comparisons between recreationally trained and highly trained individuals. The same thing can be said for Khamoui et al. (2009) and Scott & Docherty (2004). As I mentioned in the original comment, I personally think that the authors do not need to bring high-load protocol down to bring the low-load discussion. I still struggle with how relevant the discussion of fatigue here. The authors kept describing the “issues” related to high-load, but you may simply discuss the low load with BFR as an alternative approach to find the effectiveness on PAPE. Meaning that high load protocol is often used to induce PAPE (the original manuscript also discussed the high load might be preferable than low load) --> alternative approach is to use low load protocol and BFR may augment this --> here are some previous studies showing the potential benefits… This is just a suggestion but I believe that at least the citations can be cleaned up a bit.

Line 75-99. Reply to rebuttal. My intention in the original comment was not to ask to further elaborate on chronic adaptations as the current study explored the acute effects of BFR. I still believe that Lixandrao et al. (2018) is not a good representation of maximal strength adaptations between high load vs. low load BFR. Nonetheless, the bigger talking point should be why BFR may augment PAPE. The two-third of this paragraph now discusses the chronic adaptations with BFR, and no single study was brought up during the introduction to discuss PAPE on BFR. I believe that studies by Takarada et al. (2000), Laurentino et al. (2012), and Lixandrao et al. (2018) are largely irrelevant to the current study. What did a previous study show with regard to BFR on PAPE? What are the potential reasons BFR may augment PAPE? I’m aware that the authors talked more in the discussion, but I believe that the introduction can be more direct and stick with the main idea of this study.

Experimental design

Methods
Reply to rebuttal. As AOP was estimated, I think it is important to specify that it was estimated 80% AOP throughout the manuscript.
Reply: Thank you for your valuable feedback. In response to your comment, we have revised the manuscript to incorporate references to Loenneke et al.'s research. Specifically, we have used Loenneke et al. (2015) and the recent meta-analysis by Murray et al. (2021) to estimate the appropriate cuff inflation pressure based on thigh circumference.
As mentioned, please describe the estimated 80% AOP throughout the manuscript. The authors also should acknowledge the limitations of using these equations. The Loenneke’s group had a recent paper discussing this.

Line 165-170. Did the cuff remain inflated throughout the intervention? Did every participant complete the prescribed repetition scheme? This information might be useful.

Validity of the findings

Discussion
Line 293-296. Lactate itself does not cause fatigue per se, but it is now seen as a metabolic intermediate that can be used in energy substrate utilization (https://pubmed.ncbi.nlm.nih.gov/33566386/). Additionally, I have a hard time connecting the discussion about lactate and muscular adaptations with relation to mTOR signaling. What this paper discusses is the acute response to BFR, not the muscular adaptations.

Line 296-301. Similar to the comment above, I have a hard time connecting ROS and acute performance benefits with BFR. The authors still discuss long-term adaptations.

Line 302-306. “While our study’s methodology …” This sentence is very long and hard to follow. Please revise for clarity.

Line 343. Could the authors elaborate on what the “frequent application of BFR” means?

Line 344. “PNS” fatigue was never defined. Please add what it indicates.

Line 345. I think the term “long term” is not fitting here. The long term typically would imply weeks and months not minutes.

Line 351-353. “…particularly given the low-load nature of the BFR+LL protocol.” Could the authors elaborate what the statement refers to? Is it implying the increase in RFD is load dependent?

Line 359. I don’t think neural “adaptations” are fitting here (responses instead). What kind of neural responses do the authors refer to here?

Additional comments

NA

---

## Round 0.3 · Minor Revisions

Reviewer 3 has a few minor points that should be addressed in the manuscript

Reviewer 1 ·

Basic reporting

The study is well-written, using professional and technically appropriate language. However, there are some minor grammatical inconsistencies (e.g., Line 60: There is a period (.) after “…conditioning activity” that should not be present; Line 184: "participants were instructed to the Countermovement Jump (CMJ) technique" should be "participants were instructed on the Countermovement Jump (CMJ) technique"). Hyphens are still being used where endashes should be used. Endashes as used to represent a range

Raw data is shared

Experimental design

The use of blood flow restriction (BFR) in resistance training is an established area of study, and the randomized crossover design is appropriate.

Validity of the findings

No comment

Additional comments

There were detailed changes from the previous comments to the current version.

Reviewer 3 ·

Basic reporting

Abstract
Line 25. “Fifteen male strength trainers …” I think “strength trainers” can be changed to “individuals”.

Experimental design

Methods
Line 161. “Subsequently, One of the…” Please change “One” to “one”.

Line 178. “… warm-up routine, Countermovement jump …” Please change “Countermovement” to “countermovement”.

Line 184. “Subsequently, Participants were …” Please change “Participants” to “participants”.

Validity of the findings

Discussion
I am still not certain how lactate and ROS are potentiating the performance. The current narrative is largely focusing on the “recovery” aspects following exercise but recovery from exercise is not the same thing as potentiating, correct? How those metabolites may potentially “potentiate” performance is not clear to me. How confident are you on these mechanisms? Or is this largely a speculation? I think it’s okay to acknowledge that mechanisms are still unknown. I also think that you do not necessarily need to mention about the role of ROS on long-term adaptations (i.e., stick with acute responses).

Line 362. “… provide sufficient neural adaptations necessary for…” I don’t think “adaptations” is fitting here since we are discussing acute responses.

Additional comments

NA

---

## Round 0.4 · accepted · Accept

The authors addressed the reviewer comments and is ready for publication.

Reviewer 3 ·

Basic reporting

I appreciate your revisions and the effort put into addressing all the comments. All concerns have been adequately addressed, and I have no further comments. Congratulations on your work.

Experimental design

NA

Validity of the findings

NA

Additional comments

NA